


# Atmospheric sounding of the boundary layer over alpine glaciers using fixed-wing UAVs

Alexander R. Groos[1,2], Nicolas Brand[2], Murat Bronz[3], and Andreas Philipp[4]

[1]Institute of Geography, Friedrich-Alexander-Universität Erlangen-Nürnberg, 91508 Erlangen, Germany
[2]Institute of Geography, University of Bern, 3012 Bern, Switzerland
[3]Fédération ENAC ISAE-SUPAERO ONERA, Université de Toulouse, 31055 Toulouse, France
[4]Institute of Geography, University of Augsburg, 86135 Augsburg, Germany

**Correspondence:** alexander.groos@fau.de

**Abstract.**

Glaciers are an integral part of the high mountain environment and interact with the overlying atmosphere and surrounding terrain in a complex and dynamic manner. The energy exchange between the glacier surface and the overlying atmosphere controls ice melt rates and promotes the formation of a low-level katabatic jet that interacts with other, often thermally driven

winds in alpine terrain. Information on the structure of the atmospheric boundary layer over glaciers is crucial for studying the characteristics of the katabatic jet, its broader cooling effect, and its susceptibility to be broken up by strong valley or synoptic winds that promote heat advection from the ice- and snow-free periphery towards the glacier. While the number of ground-based measurements from weather stations and meteorological towers installed on glaciers for boundary layer research has increased in recent years, a lightweight and mobile measurement technique for atmospheric sounding over alpine glaciers

has not yet been available. Here we describe a new measurement technique based on a low-cost and open-source fixed-wing UAV, which allows sounding the atmospheric boundary layer over glaciers up to several hundred metres above the surface. Vertical profiles of air temperature, humidity, pressure, wind speed, wind direction and turbulence can be derived from the meteorological and flight recorder data collected by the UAV. The results of a measurement campaign on the Kanderfirn in the Swiss Alps on 16 June 2021 demonstrate the potential of the technique and highlight typical features of the boundary

layer above a melting glacier surface. The soundings reveal a persistent low-level katabatic jet, characterised by a pronounced surface-based inversion, relatively dry air, high wind speeds and enhanced turbulence, and a warmer and more humid valley wind aloft.

## 1 Introduction

Glaciers and ice caps are an integral part of the high mountain environment, interacting with the overlying atmosphere and

surrounding terrain in a complex and highly dynamic manner. The spatially and temporally highly variable energy exchange between the glacier surface and the overlying atmosphere strongly controls ice melt rates and promotes a characteristic microclimate and the formation of katabatic winds, which interact with other, often thermally driven winds in alpine terrain (e.g. Oerlemans, 2010; Farina and Zardi, 2023). A sound understanding of the principles of multi-scale glacier-atmosphere





interactions in alpine terrain and their sensitivity to climate change is essential for accurate and reliable projections of future

glacier mass loss (e.g. Mott et al., 2020; Jouberton et al., 2022; Shaw et al., 2023, 2024). Projections of the future evolution of mountain glaciers are, in turn, not only relevant in the context of sea-level rise and freshwater management (e.g. Huss and Hock, 2018; Rounce et al., 2023). They are also fundamental for assessing future local changes in the mountain-valley wind system and potential impacts on air temperature and precipitation in glacier forefields related to shrinking ice and snow cover at higher elevations (e.g. Potter et al., 2018; Salerno et al., 2023).

While the general structure of the atmospheric boundary layer over mountain glaciers and the key processes driving the local circulation in glacierised mountain terrain are relatively well understood, less is known about local and regional differences in the manifestation of katabatic winds (depending on glacier characteristics and the topographic/climatic setting) and their sensitivity to rising air temperatures in ice- and snow-free areas. Both theoretical considerations and glacio-meteorological field experiments have shown that a melting glacier surface in summer (snow or ice at 0 °C) efficiently cools the warmer air

aloft and causes a density-driven downslope flow (referred to as a katabatic wind or glacier wind), which is most developed during periods of weak synoptic flow and strong insolation (e.g. Ohata, 1989; Van Den Broeke, 1997a, b; Greuell and Böhm, 1998; Oerlemans and Grisogono, 2002). The maximum wind speed of the low-level katabatic jet is reached within a few metres above the glacier surface (Van Den Broeke, 1997b; Oerlemans, 2010; Mott et al., 2020; Nicholson and Stiperski, 2020), but the cooling effect of the katabatic wind layer can extend up to 100 m above the glacier surface (Oerlemans and Vugts, 1993;

Van Den Broeke, 1997a).

An upslope wind advecting warm and moist air from the valley usually forms during the day over the cold and dry downglacier katabatic wind (Van Den Broeke, 1997b). The glacier wind is persistent during both day and night, but can be disturbed, especially near the glacier terminus, by a strong synoptic flow or a pronounced valley wind (Oerlemans and Grisogono, 2002; Mott et al., 2020; Nicholson and Stiperski, 2020; Shaw et al., 2024). Advection of warm air from the periphery

of a glacier during disturbance of the katabatic wind layer, together with enhanced turbulent heat exchange, can locally and temporarily increase glacial melt (Mott et al., 2020; Shaw et al., 2024). This means that the response of the katabatic wind (i.e. its strengthening or weakening) to a warming environment will have a direct impact on the future energy and mass balance of mountain glaciers.

Assessing the impact of climate change on katabatic winds in glacierised mountain environments is difficult because multi-

year glacio-meteorological observations of downglacier katabatic winds and upglacier valley winds are very rare and only available from a few high elevation sites. A recent comparative study focusing on three glaciers in the Swiss Alps has shown that the near-surface cooling effect of katabatic winds (from 1 to 7 °C on warm afternoons) can vary widely between sites and depends not only on glacier size. It also depends on the resistance of a glacier to the weakening of the katabatic wind layer and the intrusion of warm air, which is controlled inter alia by the local topography and the orientation of the valley to the synoptic

flow (Shaw et al., 2024). Observations from the Himalaya suggest that the increasing difference between summer temperatures on and off glacier over the last three decades has strengthened the katabatic wind and lowered the elevation of the convergence zone between cold-dry katabatic winds and warm-moist valley winds. This effect could explain both the observed cooling and drying of proglacial areas in this region (Salerno et al., 2023). However, the continued area loss of mountain glaciers will





eventually lead to a decay of katabatic winds, as measurements from a retreating alpine glacier indicate (Shaw et al., 2023).
The decay may be further enhanced by the increase in supraglacial debris cover, which promotes the disruption of katabatic winds (Nicholson and Stiperski, 2020). In mountains that will become ice-free, warm air is likely to be advected further up the valley by local winds, as suggested by climate model simulations (Potter et al., 2018).

While meteorological measurements from automatic weather stations operated continuously or temporarily on mountain glaciers around the world have provided fundamental insights into near-surface winds, air temperature fields and turbulent
energy fluxes (e.g. Greuell and Böhm, 1998; Oerlemans and Grisogono, 2002; Shea and Moore, 2010; Petersen and Pellicciotti, 2011; Shaw et al., 2016; Steiner and Pellicciotti, 2016; Shaw et al., 2017; Mott et al., 2020; Nicholson and Stiperski, 2020; Shaw et al., 2021, 2023, 2024), techniques for atmospheric sounding to greater heights on mountain glaciers are still scarce. However, atmospheric soundings (i.e. vertical profiling of various atmospheric variables) are crucial for determining the vertical extent of the katabatic wind layer, for studying the structure of the boundary layer, and for investigating the interactions between the
katabatic wind, other thermally driven local winds (i.e. valley and slope winds), and the free atmosphere (i.e. synoptic flow) (e.g. Oerlemans, 2010). Meteorological instruments installed on a mast or tower can extend the vertical measurement range, but the logistical effort is tremendous and the maximum height is usually limited to a couple of meters above the glacier surface (e.g. Oerlemans and Vugts, 1993; Oerlemans et al., 1999; Litt et al., 2015).

Glacio-meteorological experiments with a large tethered balloon on Vatnajökull in Iceland (Oerlemans et al., 1999) and on
Pasterze in the Austrian Alps (Van Den Broeke, 1997a, b; Oerlemans and Grisogono, 2002) allowed atmospheric soundings up to several hundred metres above the glacier surface already two decades ago. However, to the authors' knowledge, since the comprehensive Pasterze experiment in the summer of 1994, no atmospheric soundings have been carried out on any mountain glacier worldwide, presumably because of the logistical challenges and high costs involved. As a low-cost and lightweight alternative to tethered balloons, Unoccupied Aerial Vehicles (UAVs) equipped with meteorological sensors have increasingly
been deployed in recent years for atmospheric boundary layer research in the high latitudes, including Antarctica, Greenland and Iceland (e.g. Reuder et al., 2009; Cassano, 2014; Jonassen et al., 2015; Cassano et al., 2016; Lampert et al., 2020; Hansche et al., 2023). However, the feasibility and suitability of UAVs for atmospheric sounding over alpine glaciers has not yet been demonstrated. Here, we describe a low-cost and open-source fixed-wing UAV for boundary layer research on mountain glaciers, present the results of a feasibility study on the Kanderfirn in the Swiss Alps in June 2021, and discuss the benefits, challenges
and limitations of using meteorological UAVs in alpine terrain.

## 2   Study area

The Kanderfirn (46.47 °N, 7.78 °E), a south-west-facing valley glacier in the Swiss Alps (see Fig. 1), was chosen as test site for the UAV-based atmospheric soundings because the setting allowed to study the influence of the alpine terrain and the extensive snow cover on the occurrence of typical local winds (i.e. katabatic, valley and slope winds) and the structure of
the atmospheric boundary layer over the glacier. Furthermore, there is already considerable experience of using UAVs on this glacier from previous campaigns (Groos et al., 2019, 2022; Messmer and Groos, 2024). The glacier currently covers an area of



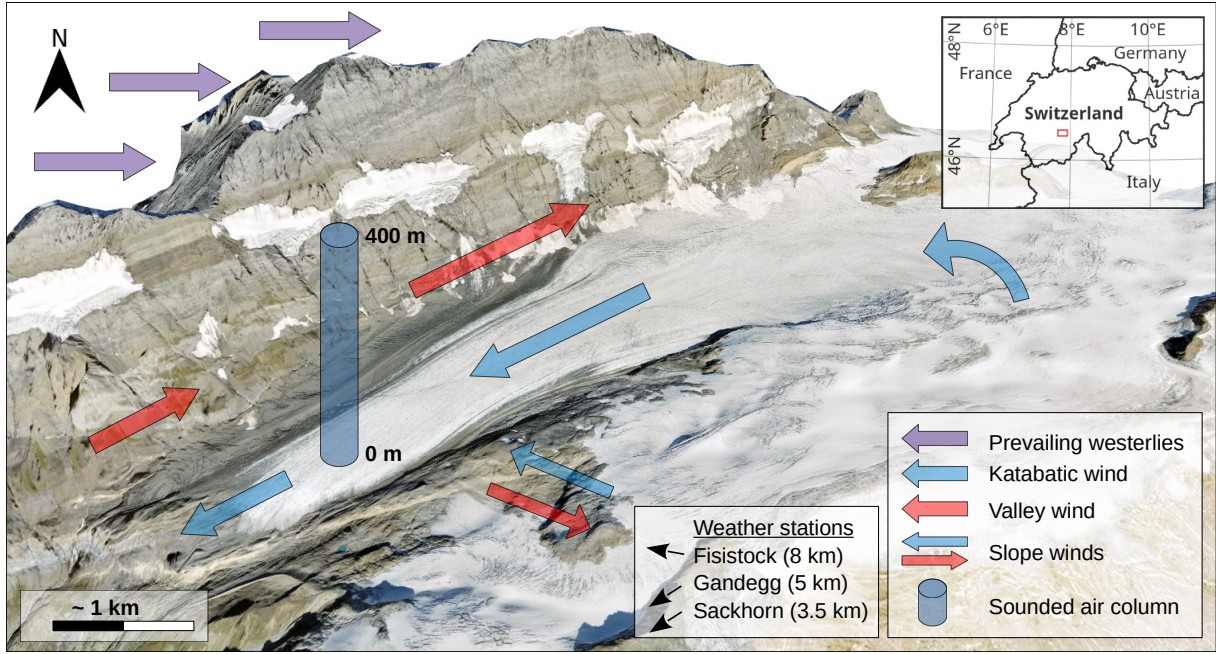

**Figure 1.** Overview of the alpine test site for the UAV-based atmospheric soundings and schematic of the synoptic flow and hypothesised dominant local winds on the Kanderfirn in the Swiss Alps. The approximate location of the nearby weather stations is indicated. The 3D model in the background is based on the SWISSIMAGE (orthophoto) and the swissALTI3D (digital surface model) from 2018, both provided by Swisstopo.

approximately 12 km$^2$ and extends from 2300 meter above sea level (m a.s.l.) at the tongue to 3200 m a.s.l. in the accumulation area. To the north, the glacier is bounded by the steep Blüemlisalp massif, with a maximum elevation of 3661 m a.s.l. (for a detailed description of the study area, see Groos et al., 2019). During the campaign on 16 June 2021, at the beginning of the

melt season, the glacier was under the influence of a pronounced anticyclone over the Sahara (see Fig. 2) and was still fully snow-covered (see Fig. 3). The snow depth in the area of the atmospheric soundings was about 2 m. Three automatic weather stations operated by the WSL Institute for Snow and Avalanche Research (SLF), one at Fisistock (46.4715 °N, 7.6739 °E; 2160 m a.s.l.), one at Gandegg (46.4293 °N, 7.7606 °E; 2720 m a.s.l.) and one at Sackhorn (46.4397 °N, 7.7662 °E; 3200 m a.s.l.), are located in the proximity of the glacier (see Fig. 1).

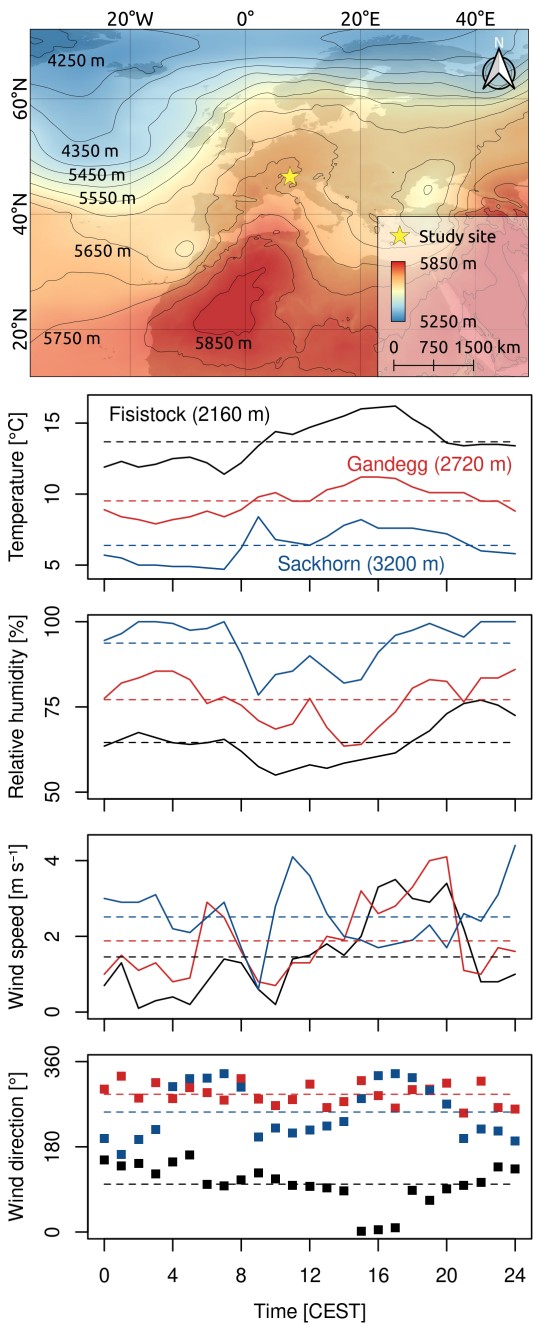

**Figure 2.** Synoptic situation at the time of the measurement campaign (16 June 2021). The synoptic weather chart shows the atmospheric thickness between the 1000 and 500 hPa pressure levels derived from ERA5 data. The study area was under the influence of an anticyclone over the Sahara. The diurnal cycle of air temperature, relative humidity, wind speed and wind direction on the day of the UAV-based atmospheric soundings is shown for three weather stations in the vicinity of the glacier (see Fig. 1).



## 3   Methodology

### 3.1   Unoccupied Aerial Systen

For the atmospheric sounding of the boundary layer over alpine glaciers, we designed a lightweight Unoccupied Aerial System (UAS) using low-cost and open-source software and hardware developed within the framework of the Paparazzi UAV project (Hattenberger et al., 2014). The UAS is very similar to the one presented by Groos et al. (2019) for photogrammetric surveys in alpine terrain and consists of a ground segment, an airborne segment (i.e. the UAV) and a communication segment. A mobile and lightweight ground control station (GCS) is necessary in the field for the configuration, monitoring and control of the UAV. Our GCS consists of a rugged outdoor laptop running the Paparazzi software, a remote control (Graupner HoTT mx-16 2.4 GHz) for manual operation of the UAV and a bi-directional wireless modem (XBee Pro S2B 2.4 GHz) for communication between the GCS and the UAV. The wireless modem supports both telemetry (downlink) and telecontrol (uplink).

The fixed-wing UAV for atmospheric sounding (see Fig. 2) was built from scratch and has a wingspan of 160 cm. We used expanded polypropylene (EPP) fuselage parts available from the aeromodelling community to build two identical flying wings. The centrepiece of the UAV is the open-source autopilot (Apogee v1.0) that supports automatic and autonomous flight (Hattenberger et al., 2014). It was developed as part of the Paparazzi UAV project and can be replicated using the hardware design published online (https://wiki.paparazziuav.org/wiki/Apogee/v1.00). A rear-mounted brushless electric motor (Hacker A30-12XL V4) with carbon folding propellers in pusher configuration provides the necessary thrust. Two digital servomotors (KST DS145 MG) control the ailerons. For automatic flight and communication with the GCS and remote control, the UAV is equipped with a Global Navigation Satellite System (GNSS, Drotek U-blox NEO-M8T), a wireless modem (XBee Pro S2B 2.4 GHz) and a receiver (Graupner HoTT mx-16 2.4 GHz). A 5000 mAh lithium polymer battery powers the whole system and supports flight times up to 45 minutes. Several small lights on the UAV enable atmospheric sounding at night. A digital humidity and temperature sensor (Sensiron SHT75), with an accuracy of $\pm 1.8$ % for relative humidity and of $\pm 0.3$ °C for air temperature (Sensirion, 2024), is connected to the autopilot. The sensor is housed in a white tube to protect it from direct solar radiation (see Fig. 2). During the flight, air flows through the tube from the front to the back, ensuring adequate ventilation of the sensor. All collected data is stored on an SD card. In total, the UAV weighs less than 2 kg and costs about 1200 EUR (excluding the GCS and remote control).

To cross-calibrate the temperature and humidity sensors on both UAVs (A2NO1 III and A2NO1 IV), a 24-hour intercomparison measurement was performed indoors from 4 to 5 June 2021 with an independent reference logger (see Fig. A1). A fan in front of the two UAVs and reference logger ensured adequate ventilation of the sensors. The mean difference in air temperature between both SHT75 sensors was 0.43 °C and the mean difference in relative humidity was 1.5 % (see Fig. A2).

### 3.2   Atmospheric soundings

The air column sounded by fixed-wing UAV on 16 June 2024 was located in the central lower part of the glacier (46.46989 °N, 7.777853 °E, 2430 m a.s.l.), about 1 km from the terminus (see Fig. 1). A total of 8 flights, each consisting of two vertical profiles, were performed between 10 am and 5 pm Central European Summer Time (CEST) (see Table 1). Take-off and





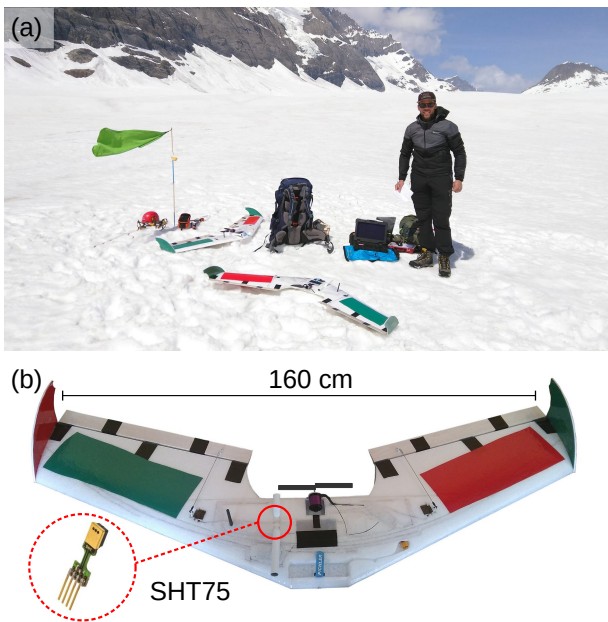

**Figure 3.** UAS and local conditions during the measurement campaign. (a) The Kanderfirn was completely snow-covered. The provisional weather vane (green flag) indicates the cold and dry katabatic wind that prevailed during the entire campaign. (b) Close-up view of the low-cost and open-source fixed-wing UAV equipped with a temperature and humidity probe (SHT75).

landing of the fixed-wing UAV on the glacier was performed manually. Planned take-offs on the hour had to be delayed at certain times of the day due to poor GPS accuracy (see Table 1). For the vertical soundings, the UAV automatically followed a
pre-programmed flight plan to measure air temperature and relative humidity up to 400 m above the glacier surface (see Fig. 4). After a rapid ascent, the UAV circled at a pre-defined maximum height (see Table 1) for 60 seconds to give the SHT75 sensor enough time to adapt to the ambient air. The descent was performed in a spiral with a radius of 75 m to support the derivation of wind speed and wind direction from the GNSS data (see Section 3.3). The sink rate was low to minimise the effect of sensor inertia (5 s for air temperature and 8 s for relative humidity to adapt to 63 % of a signal change). Once the UAV reached a
height of less than 10 m above ground level (m a.g.l.), the sounding was repeated. A complete sounding (consisting of two vertical profiles) took about 10 to 15 minutes.

### 3.3   Data processing and analysis

The meteorological data (air temperature, relative humidity and air pressure) and flight recorder data (e.g. roll, pitch, yaw, battery voltage) collected during each sounding are stored in two different log files. While the meteorological data are stored
in a human readable text file, the flight recorder data are stored in a binary file that is decoded using checksums for further analysis. Decoding, post-processing and reformatting of the data is performed by the FORTRAN software package mmp (mobile measurement post-processing) by Philipp (2024), which assigns all sensor messages to unified time steps and detects





**Table 1.** Key figures of the UAV-based atmospheric soundings on 16 June 2021.

| UAV name | No. of profiles | Takeoff time | Landing time | Max. height (m a.g.l.) |
|---|---|---|---|---|
| A2NO1 III | 2 | 10:41 | 10:51 | 250 |
| A2NO1 III | 2 | 11:10 | 11:22 | 350 |
| A2NO1 III | 2 | 12:00 | 12:12 | 350 |
| A2NO1 IV | 2 | 13:24 | 13:41 | 350 |
| A2NO1 IV | 2 | 14:01 | 14:17 | 350 |
| A2NO1 IV | 2 | 15:07 | 15:23 | 400 |
| A2NO1 IV | 2 | 16:00 | 16:15 | 400 |
| A2NO1 IV | 2 | 16:45 | 17:00 | 400 |

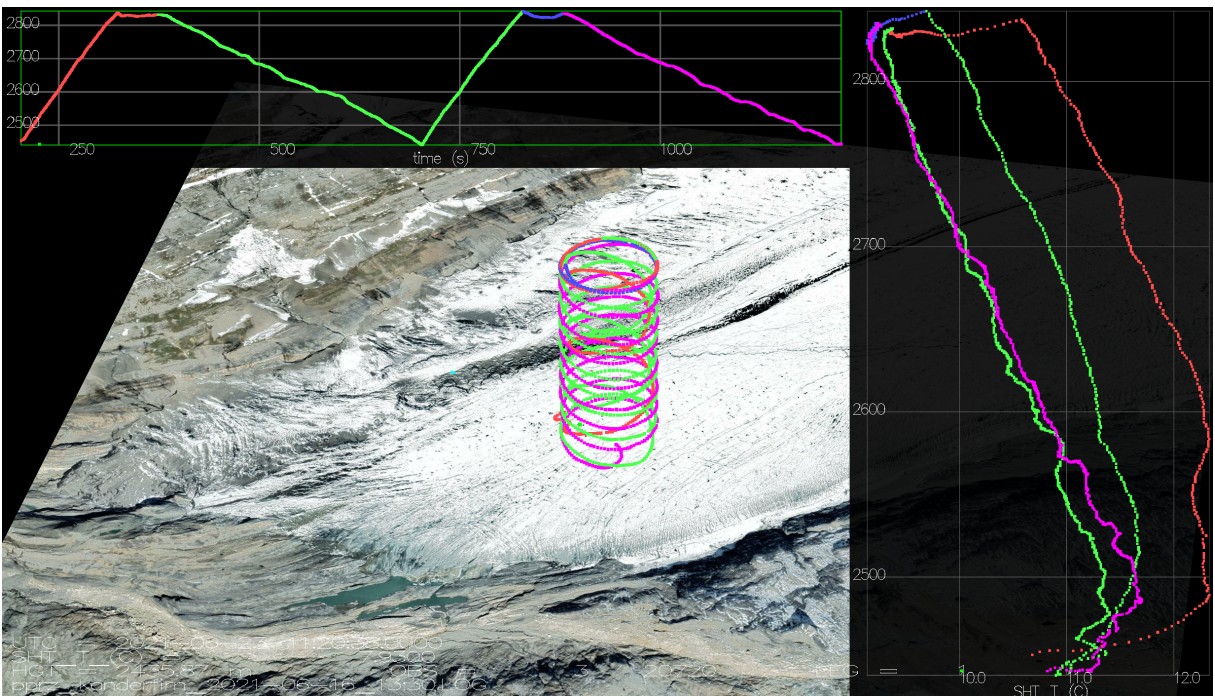

**Figure 4.** Screenshot of the graphical user interface of the open-source software package mmp for data post-processing and 3D visualisation. The top panel shows the flight altitude (m) over time, the middle panel shows the flight path of two subsequent soundings and the right panel shows the temperature profiles for two climbs and two descents at about 13:30 CEST. Note that the air temperature measured during the two rapid climbs has a warm-bias due to the inertia of the sensors.

take-offs, touch-downs and outliers in the messages from the GNSS module. Time lag correction for the SHT75 sensor and calculation of horizontal wind components can also be performed by this package.





For the analysis of vertical and temporal variations of air temperature, humidity, wind speed, wind direction and turbulence over the glacier surface, only the data from the two consecutive descents (i.e. downward spirals) of each flight were considered, because the measurements from the two ascents are strongly affected by the rapid rate of climb (see Fig. 4). The first descent of each sounding lacks data from the lowest metres above the surface, as the uneven terrain and GPS accuracy did not support flying below 10 m a.g.l. before the start of the second climb. Due to the limited GPS accuracy, the recorded flight altitude varied

by a few metres between the different soundings. Therefore, we used the average altitude before take-off and after landing of each flight to standardise the base height of all soundings.

    For each individual vertical profile on 16 June 2021, air temperature was aggregated into 1 m height intervals for consistent comparison of the different soundings by applying a central moving average with a sampling window of 5 m (2 m above and 2 m below each 1 m layer) to the original measurements. The air temperature measured by UAV A2NO1 III during the first

three soundings (see Table 1) was adjusted to harmonise the SHT75 sensors of both UAVs (A2NO1 III and A2NO1 IV) using a correction factor of 0.43 °C obtained from the 24-hour intercomparison measurement (see Appendix A). To estimate the thickness of the cool katabatic wind that prevailed throughout the campaign, we calculated the top height of the surface-based inversion (SBI) visible in all air temperature profiles as follows. We considered air temperature ($T$) as a function of altitude ($z$):

$$T = T(z) \tag{1}$$

To eliminate the short-term variability superimposed on the general air temperature change with altitude, we applied a low-pass filter (local polynomial regression) to the air temperature measurements from the second descent, which went down to the glacier surface. We then computed the first derivative of the smoothed air temperature profile with respect to altitude because the top height of the SBI is characterised by a transition of the temperature change rate from positive to negative values:

$$T'(z) = \frac{dT}{dz} \tag{2}$$

The lowest altitude ($z = z_i$) where the first derivative is zero was determined as the top height ($z_i$) of the SBI:

$$z_i = \min\{z \mid T'(z) = 0\} \tag{3}$$

To calculate the lapse rate (i.e. air temperature gradient) above and below the top height of the SBI ($z_i$), we performed a simple linear regression (see Appendix B). For the data points below $z_i$, the linear regression model can be expressed as:

$$T(z) = a_1 z + b_1 \quad \text{for} \quad z \leq z_i, \tag{4}$$

where $a_1$ is the lapse rate below $z_i$ and $b_1$ is the intercept. For the data points below $z_i$, the linear regression model can be expressed as:

$$T(z) = a_2 z + b_2 \quad \text{for} \quad z \geq z_i, \tag{5}$$

where $a_2$ is the lapse rate above $z_i$ and $b_2$ is the intercept. In addition to the UAV-based lapse rates, we also calculated

the environmental lapse rate for the study area at the time of each sounding using data from three nearby weather stations: Fisistock, Gandegg and Sackhorn (see Appendix B).



Since spatio-temporal variations in relative humidity ($RH$) can be the result of changes in air temperature, air pressure and/or water vapour, it is not an unambiguous parameter for analysing and interpreting changes in air moisture. Specific humidity ($q$) is more suitable as it is not sensitive to air temperature or pressure. We have therefore converted the recorded relative humidity (%) into specific humidity (g/kg) as follows:

The partial vapour pressure at saturation ($p_s$) for a given air temperature ($T$) is calculated using empirical constants from DWD (1976) by:

$$p_s(T) = E_0 \left( \frac{A \cdot T}{B + T} \right), \tag{6}$$

where $E_0$ = 6.10780 hPa, A = 17.08085 and B = 234.175 $K$. The actual vapour pressure ($e$) is then given by:

$$e = p_s \frac{RH}{100}, \tag{7}$$

Finally, the specific humidity ($q$) in g per kg is calculated as:

$$q = \frac{m_v}{m_a} \left( \frac{e}{p - 0.377e} \right) 1000, \tag{8}$$

where $m_v$ is the molar mass of water vapour (18.01534 g/mol), $m_a$ is the molar mass of dry air (28.9644 g/mol) and $p$ is the pressure (in hPa) either measured or calculated from elevation assuming the standard atmosphere. As with air temperature, specific humidity was then resampled at 1 m height intervals using a central moving average to ensure consistent comparison of the different soundings.

Unlike air temperature and humidity, wind speed, wind direction and turbulence were not measured directly. These three parameters had to be derived from the GNSS data and the recordings made by the inertial measurement unit. The wind estimation algorithm is based on the concept described by Mayer et al. (2012) and Bonin et al. (2013). The wind direction is estimated from the variation in ground speed during a full circle of the flight path with all UAV flight controls, in particular throttle and pitch, held constant. The (opposite) wind direction is then the flight direction at maximum (minimum) ground speed within a circle. The wind speed can be assumed to be the difference between the average ground speed during a circle and the maximum or minimum ground speed.

As a rough approximation of turbulence, the roll rate recorded by the inertial measurement unit at 4 Hz can be used, since eddies with diameters ranging from a few decimetres to a few metres, acting differently on the two wings, cause short rotational movements around the longitudinal axis of the UAV. Intensity of turbulence can then be estimated by calculating the standard deviation of the roll rates within a certain vertical section. In order to focus on the vertical scale of several meters, the turbulence proxy is calculated for 1 m height intervals by applying a central moving standard deviation with a sampling window of 10 m to the recorded roll rate. This proxy measure of turbulence can only be a rough estimate as large parts of the turbulence spectrum are missing. Therefore absolute values e.g. for turbulence kinetic energy (TKE) can not be derived. However, it is able to depict the general tendency of relatively increasing or decreasing turbulence. Details on this method are published in a separate paper.





## 4 Results

The atmospheric boundary layer above the glacier tongue of Kanderfirn warmed steadily from late morning (10:45) to late afternoon (16:05) on 16 June 2021, with a particularly strong warming around noon (12:05) (see Fig. 5). Interestingly, none of the three nearby weather stations recorded any particular warming around midday, which could indicate the advection of warm air (see Fig. 2). During the last sounding of the campaign in the late afternoon (16:45), a cooling of the whole air column from the glacier surface up to 400 m a.g.l. had already started. While the maximum air temperatures during each sounding were reached a few tens of meter above the ground, minimum air temperatures were recorded close to the glacier surface (see Fig. 5).

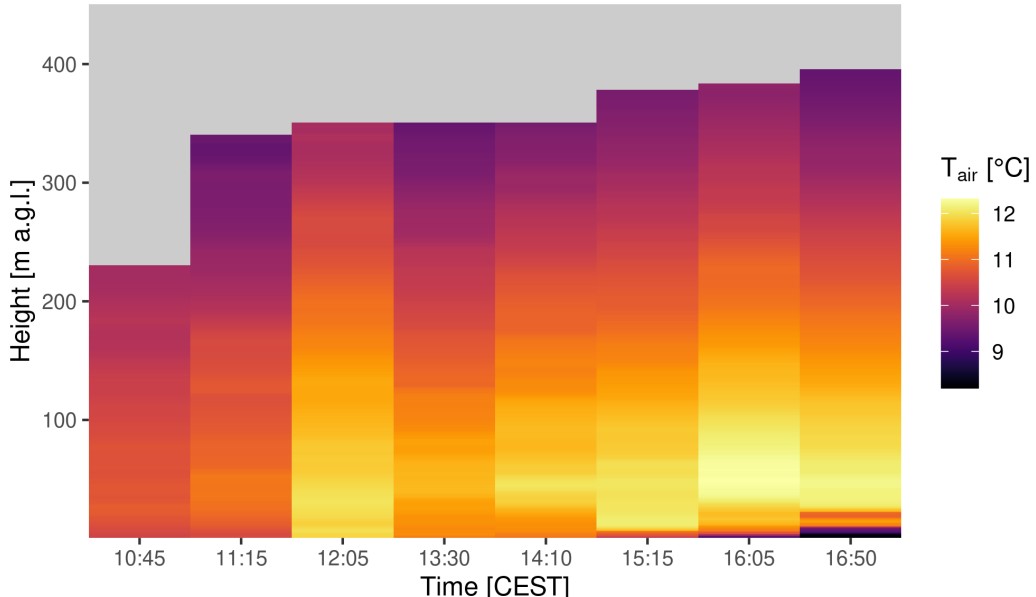

**Figure 5.** Course of the air temperature over the tongue of the Kanderfirn on 16 June 2021 derived from the second descent of each UAV flight.

In addition to the general course of air temperature over the glacier surface, which is well visible in the heat map (Fig. 5), the air temperature profiles up to 400 m a.g.l. derived from the two consecutive UAV descents of each sounding show characteristic variations in the vertical structure of the atmospheric boundary layer (Fig. 6). The two profiles from each sounding are more or less consistent (apart from deviations in the lowest metres of the 16:05 profiles), which gives confidence in the reliability and representativeness of the individual soundings. A surface-based inversion (SBI), most developed in the afternoon hours, can be seen in every second descent profile (reaching the ground) and in most first descent profiles (reaching about 10 m a.g.l.). The top height of the SBI varied between 10 and 50 m a.g.l. during the campaign (Fig. 7a). There was a marked increase in air temperature from the glacier surface to the top of the SBI. The warming rate (equivalent to a negative lapse rate) from the


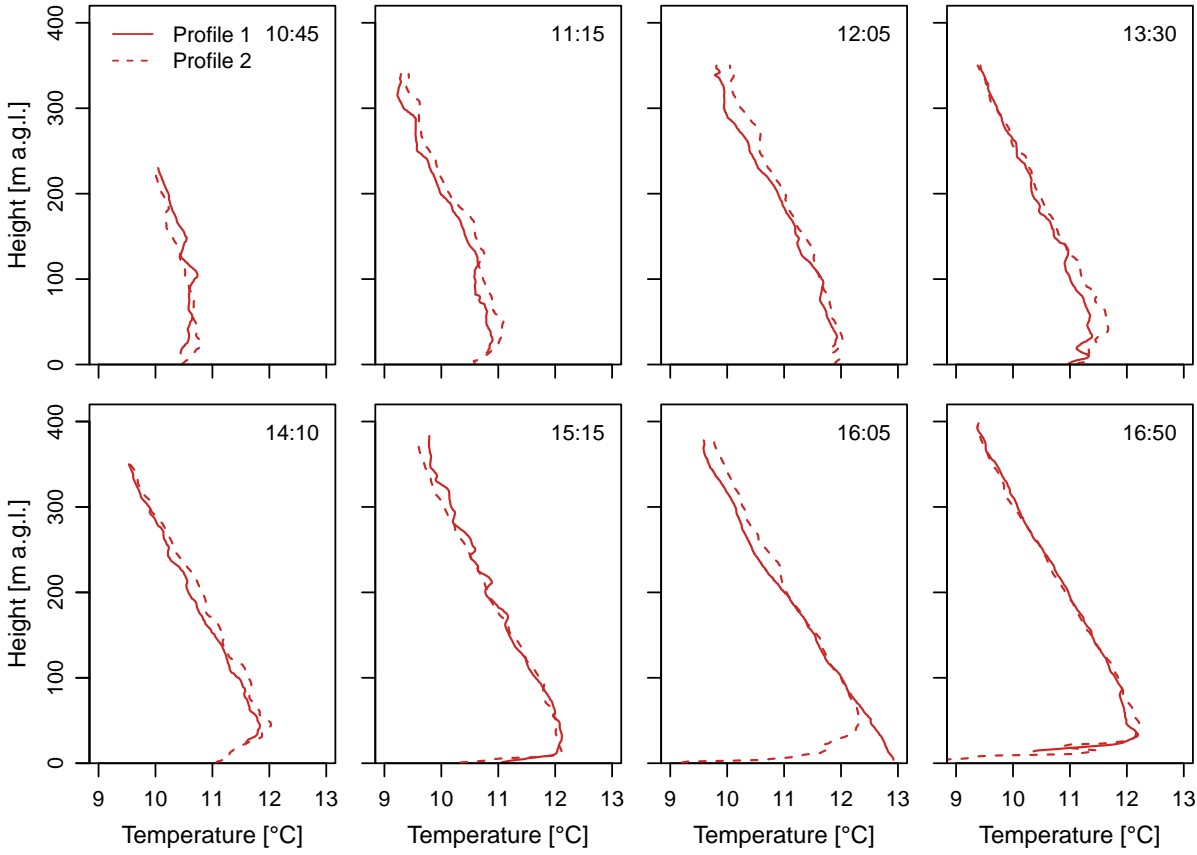

**Figure 6.** Air temperature profiles over the tongue of the Kanderfirn on 16 June 2021. Note that only the measurements from the two consecutive UAV descents (not from the ascents) are shown here.

glacier surface to the top of the SBI ranged from 0.02 °C per 10 m (at 12:05) to 1.8 °C per 10 m (at 15:15) and had a mean value of 0.5 °C per 10 m (Fig. 7c). Above the SBI, the air temperature decreased linearly up to the maximum height of the

sounded air column. The lapse rate above the SBI increased from 0.4 °C per 100 m in the late morning to about 0.8 °C per 100 m after noon (Fig. 7b), due to the stronger warming rate of the lower part of the boundary layer above the SBI (Figs. 5 and 6). Compared to the UAV-derived lapse rates, the environmental lapse rates calculated from the data of the three nearby weather stations show almost no temporal variability and are slightly higher on average (0.78 vs. 0.71 °C per 100 m). Overall, the air temperature change rate below the top height of the SBI is 15 times greater than above.

Similar to the air temperature, the water vapour in the sounded air column above the glacier tongue increased steadily during the campaign, from about 7.5 g kg$^{-1}$ (vertical mean) in the late morning to about 8.5 g kg$^{-1}$ (vertical mean) in the late afternoon (see Fig. 8). The heat map clearly shows that the cool air below the top height of the SBI is significantly drier than the air above. For example, in the late afternoon (at 16:05 and 16:45), specific humidity increased from less than 6.5 g kg$^{-1}$ near the

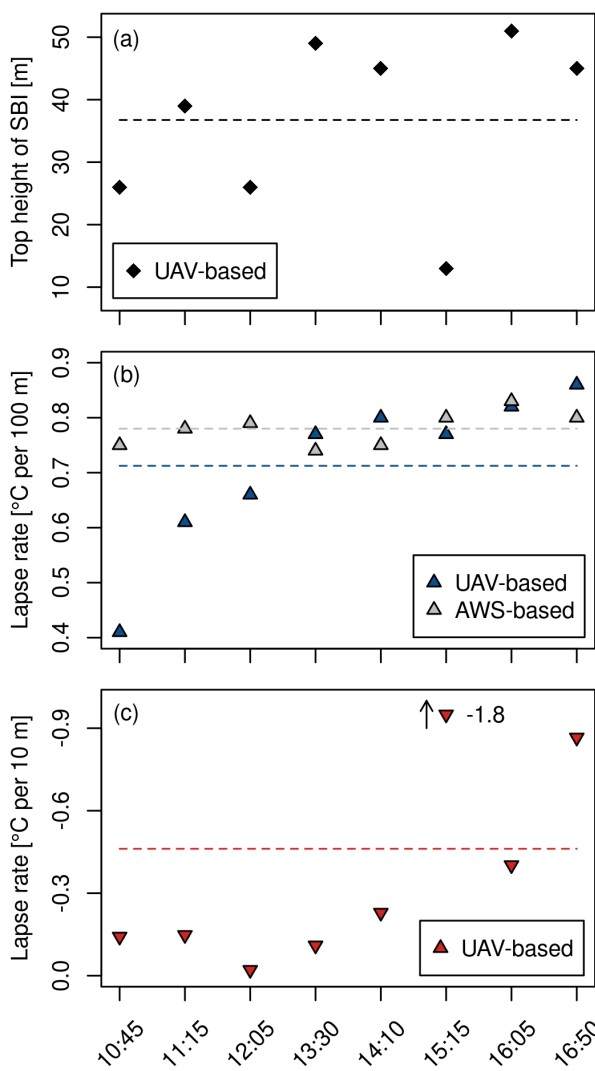

**Figure 7.** Course of the structure of the atmospheric boundary layer over the tongue of the Kanderfirn on 16 June 2021. (a) Top height of surface-based inversion. (b) Lapse rate above the surface-based inversions (SBIs) in °C per 100 m derived from the UAV soundings and environmental lapse rate in the area of the glacier calculated from data of three nearby weather stations (Fisistock, Gandegg and Sackhorn). (c) Lapse rate below the SBIs in °C per 10 m. Dashed lines show the mean.




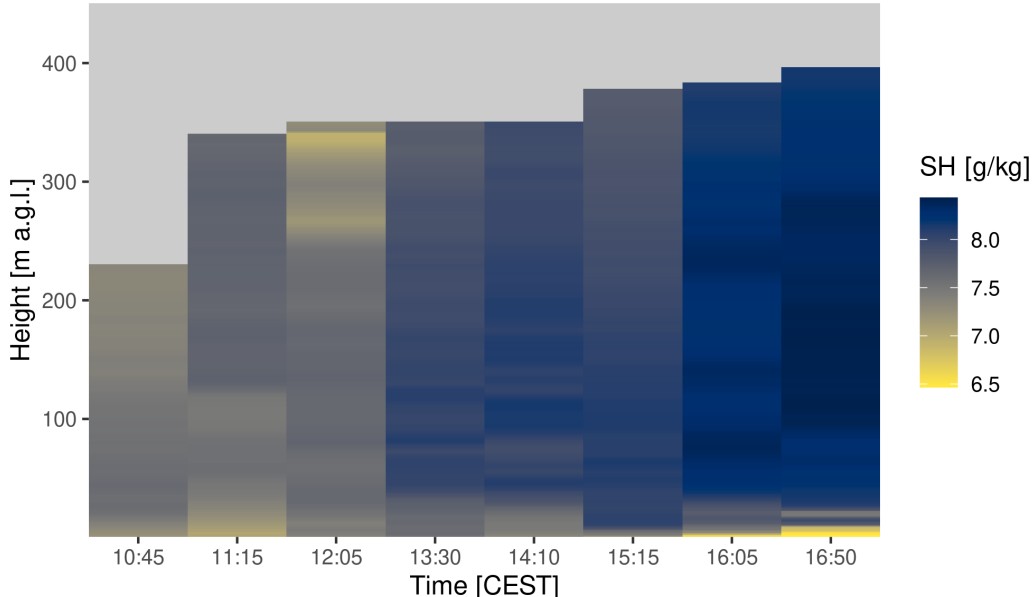

**Figure 8.** Course of the specific humidity (SH) over the tongue of the Kanderfirn on 16 June 2021 derived from the second descent of each UAV flight.

glacier surface to almost 8.5 g kg$^{-1}$ at less than 50 m height. Apart from this, water vapour generally varied much less with
altitude than air temperature (Fig. 9).

In contrast to air temperature and water vapour, wind speed did not increase during the campaign and was highly variable
vertically and temporally. Within individual profiles, wind speed varied from less than 1 m s$^{-1}$ to a maximum of 8 m s$^{-1}$ (Fig.
10). The wind speed profiles from the two consecutive descents of each flight are in relatively good agreement, but also show
short-term deviations at similar altitudes (Fig. 11). Apart from the soundings at 10:45 and 13:30, a clear pattern is visible in all
profiles that reached the ground: high wind speeds up to 8 m s-1 were observed close to the glacier surface. From the glacier
surface to about 100-200 m a.g.l. the wind speed decreased to 0-2 m s$^{-1}$. Above the calm layer, the wind speed increased again,
reaching values of 3-6 m s$^{-1}$ at about 400 m a.g.l. (Fig. 11). The mean wind speed (about 2-4 m s$^{-1}$) measured at the three
nearby weather stations during the campaign (Fig. 2) was much lower than the wind speed observed near the glacier surface
(cf. Fig. 11).
The dominant wind direction in the lowest 100-150 m of the sounded atmospheric boundary layer above the glacier surface
was northeast (about 55°; see Fig. 12), similar to the orientation of the lower part of the glacier and the assumed direction of
the glacier wind (cf. Fig. 1). In contrast, the main wind direction above 250 m a.g.l. was southwest (about 235°; see Fig. 12),
similar to the orientation of the ice-free valley below the glacier terminus and the assumed direction of the valley wind (cf.
Fig. 1). The layer of strong wind shear of the order of 180° (from northeast to southwest) varied between the soundings from
about 50-150 m a.g.l. (e.g. at 16:05; see Fig. 12) to 150-250 m a.g.l. (e.g. at 12:05; see Fig. 12) and corresponded well with





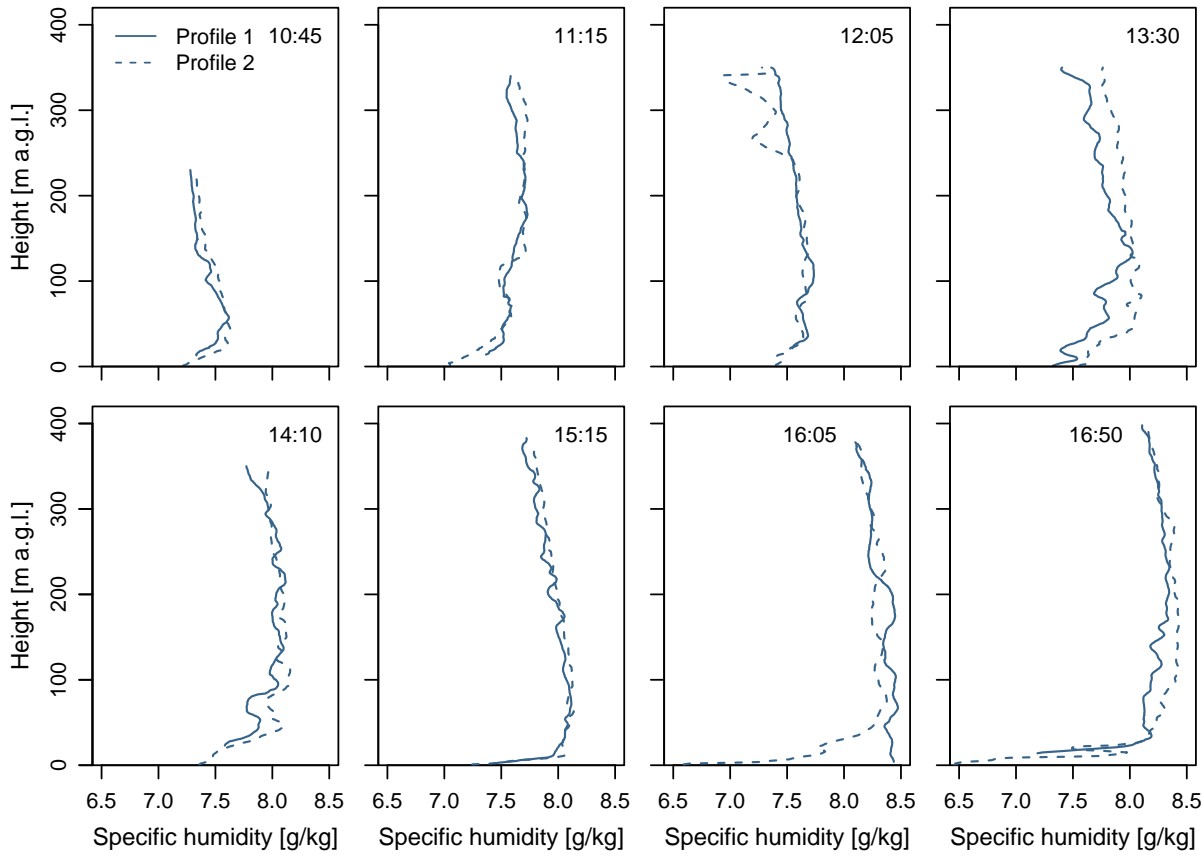

**Figure 9.** Humidity profiles over the tongue of the Kanderfirn on 16 June 2021. Note that only the measurements from the two consecutive UAV descents (not from the ascents) are shown here.

the layer of reduced wind speed (cf. Fig. 11). A clear difference in wind direction can also be seen in the data from the three nearby weather stations. While the two upper stations at 2720 and 3200 m a.s.l. recorded a mean wind direction of 290° and 255° (west) on 16 June 2021, the lower station at 2160 m a.s.l. at the end of the valley below the Kanderfirn recorded a mean wind direction of 100° (east), pointing towards the glacier terminus (Fig. 2).

The simple turbulence proxy derived from the UAV roll rate (the standard deviation at 10 m height intervals) revealed short-term and small-scale variations in turbulence over the glacier during the campaign (Fig. 13). In general, turbulence was most pronounced in the lower 30 m of the atmospheric boundary layer. Increased turbulence was also observed in the layer of wind shear and decreased wind speed. For example, during the last sounding (16:50), wind speed decreased from the glacier surface to about 200 m a.g.l. (Fig. 11) and wind shear (from northeast to southwest) occurred between about 100 and 250 m a.g.l.

(Fig. 12). Exactly in this layer the turbulence was increased compared to the air layer below (30-100 m a.g.l.) and above (250-400 m a.g.l.). Consequently, during the sounding at 13:30, when no clear wind shear and no distinct wind speed change was observed, no increased turbulence could be detected (except from the lowest 20 m) (Fig. 13).

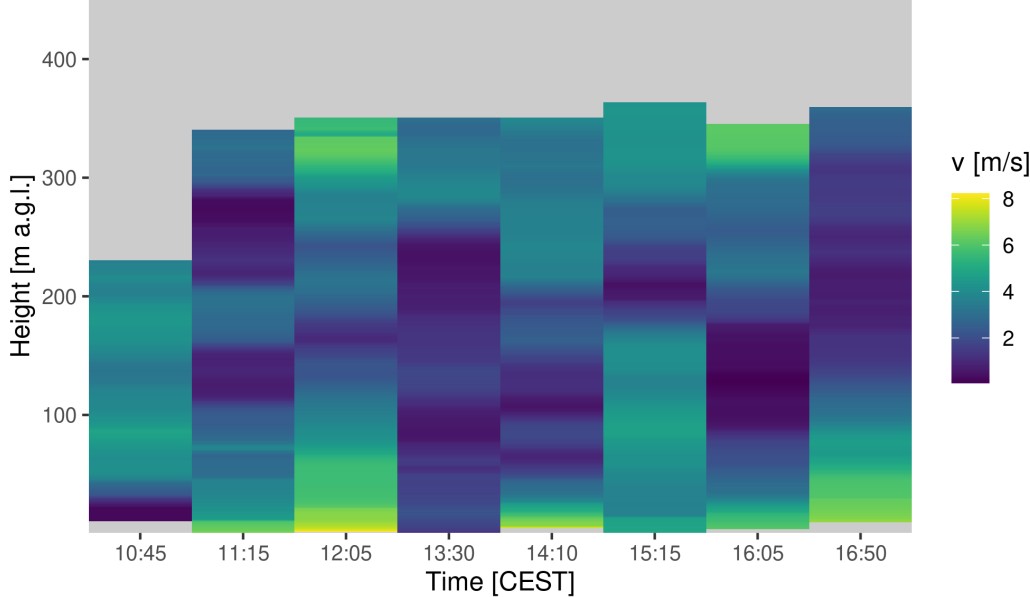

**Figure 10.** Course of the wind speed ($\nu$) over the tongue of the Kanderfirn on 16 June 2021 derived from the second descent of each UAV flight.

## 5    Discussion

The aim of the following discussion is (i) to outline the prospects and challenges of UAV-based atmospheric sounding in alpine terrain and (ii) to elaborate on the insights that the described measurement technique can provide into the local mountain-valley wind circulation and the structure of the atmospheric boundary layer over alpine glaciers.

### 5.1    Atmospheric sounding with UAVs in alpine terrain

The feasibility study and measurement campaign on the Kanderfirn in the Swiss Alps on 16 June 2021 has demonstrated the suitability of the developed low-cost and open-source fixed-wing UAV for atmospheric sounding of the boundary layer over alpine glaciers. No major technical problems were encountered during the campaign. However, the campaign revealed some practical challenges and technical limitations that should be considered for future applications and the further development of the sounding technique presented. Manual take-off (i.e. hand launch) of fixed-wing UAVs above 2300 m a.s.l. is generally possible if a headwind, such as the persistent katabatic wind during the campaign, provides sufficient uplift. The downsite is that calm conditions increase the potential risk of a crash during launch at this altitude. In addition, hand launches require considerable experience and carry an inherent risk of injury. A much safer and more reliable technique for launching fixed-wing UAVs, which we have relied on in all subsequent campaigns to date, is the use of a bungee rope anchored to the glacier with an ice screw. The bungee rope is attached to a pin that is inserted into a small tube at the underside of the UAV and is





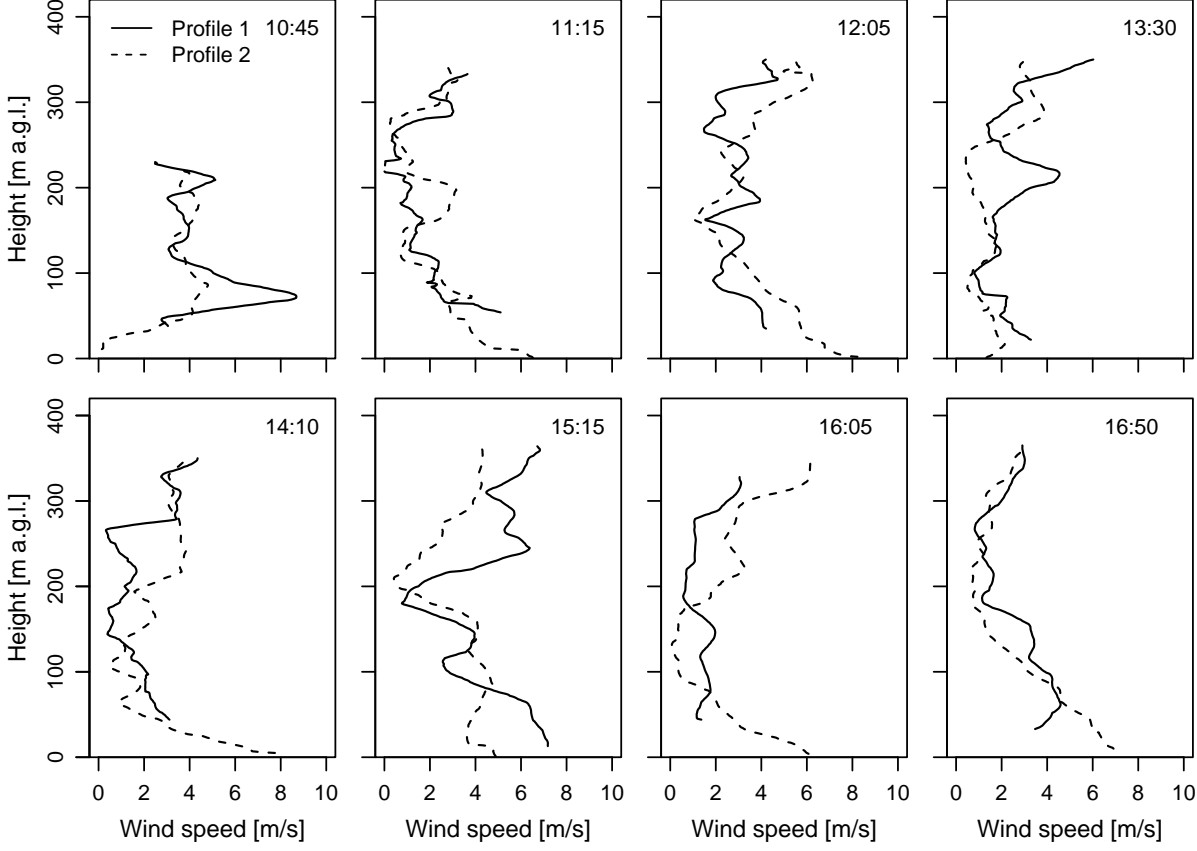

**Figure 11.** Wind speed profiles over the tongue of the Kanderfirn on 16 June 2021. Note that only the measurements from the two consecutive UAV descents (not from the ascents) are shown here.

automatically released when the UAV passes the anchor point. Due to the rough surface and presence of crevasses, moulins and meltwater channels, automatic landing of a fixed-wing UAV on a glacier is very difficult. Manual landing is possible, but
requires extensive training in the operation of fixed-wing UAVs.

Operating a fixed-wing UAV with a relatively large wingspan of 160 cm in alpine terrain has both advantages and disadvantages. Obviously, carrying a fixed-wing UAV of that size in alpine terrain is not convenient. However, the design of the fixed-wing UAV supports gliding and the operation at higher altitudes where air density is significantly reduced. Moreover, the large and coloured surface makes it possible to monitor the UAV at greater heights (up to several hundred metres above
ground). This is important for safety reasons. In mountain ranges with heavy helicopter traffic, such as the Alps, automatic UAV operations may require manual intervention and sudden landings. In addition, UAV operations beyond visual line of sight are prohibited by most aviation authorities unless an exemption has been granted (EASA, 2023). Atmospheric soundings up to several hundred metres above ground are no longer possible in the Alps without special permission since the publication of the new EU drone regulations in January 2021, which limit the maximum flight height to 120 m a.g.l. (note that the EU





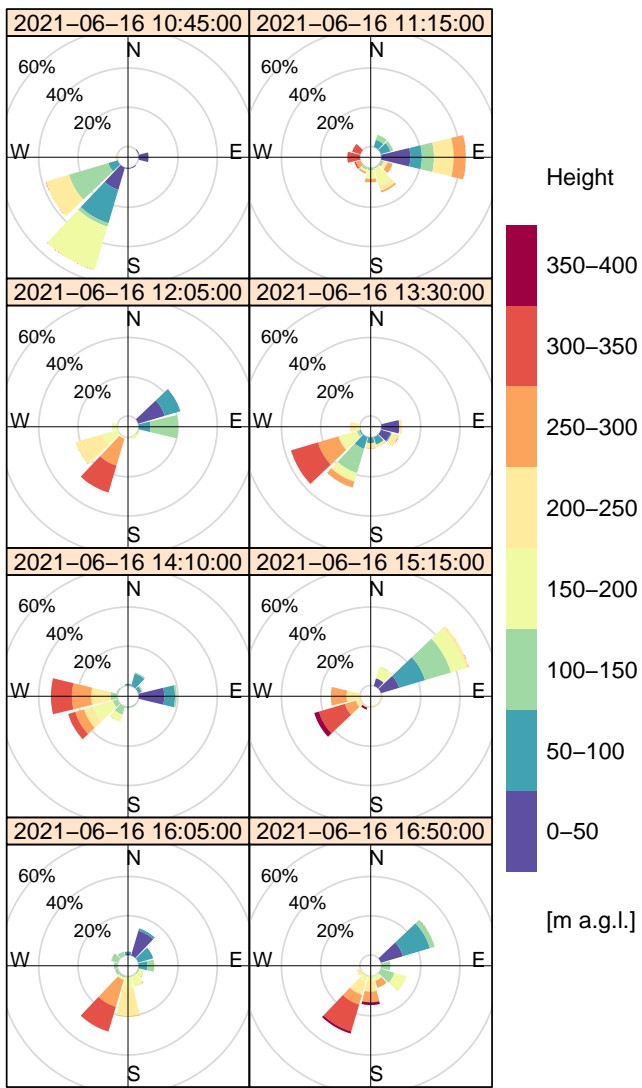

**Figure 12.** Wind direction at different heights above the tongue of the Kanderfirn on 16 June 2021 derived from the second descent of each UAV flight.





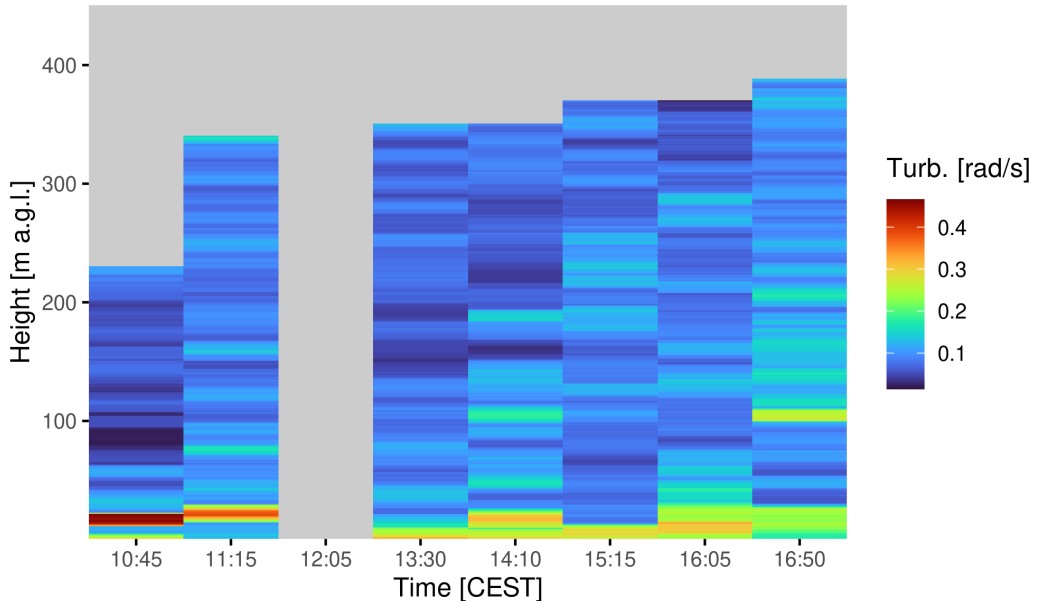

**Figure 13.** Course of the turbulence proxy over the tongue of the Kanderfirn on 16 June 2021 derived from the second descent of each UAV flight. The flight recorder file from the 12:05-flight was corrupt and could therefore not be analysed.

drone regulations were not yet in force in Switzerland during the campaign on the Kanderfirn in June 2021). Sounding the lowest 10-15 m a.g.l. is not practical with fixed-wing UAVs, as safe operation over rough surfaces cannot be guaranteed in either manual or automatic mode. In an ideal setup, atmospheric sounding with fixed-wing UAVs would be complemented by ground-based measurements (i.e. weather station and/or meteorological tower) and quadcopter soundings for the lowest tens of metres.

Compared to rotary-wing UAVs, where platform-induced heating can bias temperature measurements below the vehicle due to rotor downwash (e.g. Greene et al., 2018), the temperature and humidity sensor in the tube on the presented fixed-wing UAV is protected from solar radiation and major heat sources, and is naturally ventilated during flight. However, low ascent and descent rates are recommended to account for the effect of sensor inertia on air temperature and humidity measurements. Otherwise, the sensor time lag must be corrected (Reuder et al., 2009). While off-the-shelf UAVs usually do not support the (easy) integration of scientific sensors and rather serve as a mobile platform for stand-alone sensors (e.g. Hansche et al., 2023; Messmer and Groos, 2024), UAVs tailored for scientific purposes, such as the presented fixed-wing UAV, support the integration of different sensors and allow the combined analysis of meteorological and flight recorder data (Reuder et al., 2009, 2012; Cassano et al., 2016). Flight recorder data are beneficial because they provide additional information to estimate atmospheric parameters such as wind speed, wind direction and turbulence, which go beyond the classical measurements of air temperature and relative humidity (Mayer et al., 2012; Reuder et al., 2012; Cassano et al., 2016). However, a drawback of the presented sounding technique, which prevents easy replication, is the extensive training required to safely fly and land a





fixed-wing UAV in alpine terrain. A more user-friendly system, especially for soundings in complex terrain, are hybrid UAVs that combine efficient forward flight with hovering and vertical take-off and landing capabilities. Hybrid UAVs for scientific research are currently under development, but still need to be tested in high mountain environments (Smeur et al., 2019; Bronz et al., 2020).


## 5.2   Structure of the atmospheric boundary layer over alpine glaciers

The vertical profiles of air temperature, specific humidity, wind speed, wind direction and turbulence from Kanderfirn add to the sparse global dataset of atmospheric soundings over glacierised terrain and provide detailed insights into the structure of the boundary layer over an alpine glacier during the early phase of the ablation season. Apart from the Kanderfirn campaign,

meteorological measurements with tethered balloons and UAVs up to a few hundred metres above the glacier surface are available from only four sites worldwide: From a glacio-meteorological field experiment on Pasterze in the Austrian Alps in summer 1994 (Van Den Broeke, 1997a, b; Oerlemans and Grisogono, 2002), from a glacio-meteorological field experiment on the Vatnajökull ice cap in Iceland in summer 1996 (Oerlemans et al., 1999), from a meteorological campaign on the Hofsjökull ice cap in Iceland in summer 2007 (Reuder et al., 2009, 2012), and from a field campaign on Mittivakkat Gletsjer in southeast

Greenland in summer 2019 (Hansche et al., 2023). Although the topographical and climatic conditions vary greatly between the five sites, the soundings from the different campaigns reveal some general characteristics of the structure of the atmospheric boundary layer over glaciers and ice caps.

A key feature of the atmospheric boundary layer over mountain and outlet glaciers during the ablation season is the development of a cool and persistent density-driven katabatic wind. Katabatic winds have been observed during all of the summer

field campaigns mentioned above, and are also evident in data from numerous on-glacier weather stations (e.g. Petersen and Pellicciotti, 2011; Petersen et al., 2013; Mott et al., 2020; Nicholson and Stiperski, 2020; Shaw et al., 2023, 2024). While measurements within the lowest metres above the glacier surface are crucial for investigating turbulent energy fluxes and determining the height of the maximum wind speed of the low-level katabatic jet (Van Den Broeke, 1997b; Oerlemans, 2010; Mott et al., 2020; Nicholson and Stiperski, 2020), the vertical extent of the katabatic wind layer cannot be determined with

weather stations and meteorological towers. The tethered balloon and UAV soundings show that the katabatic wind layer is characterised by a pronounced surface-based inversion up to several tens of metres above the glacier surface (Van Den Broeke, 1997a; Oerlemans et al., 1999). While the maximum extension of the cooling effect of the katabatic wind layer has not been specifically investigated in previous studies (cf. Van Den Broeke, 1997a; Oerlemans et al., 1999; Hansche et al., 2023), the results from the Kanderfirn campaign show clearly that the top height of the surface-based inversion varies considerably in time

and that the cooling effect at this location can reach up to 50 m above the glacier surface (cf. Fig. 7). The maximum cooling observed in the lowest 50 m of Kanderfirn was of the order of 4 °C in the late afternoon (cf. Fig. 6). This is in agreement with theory, which predicts stronger katabatic winds for higher air temperatures outside the glacier (Ohata, 1989; Oerlemans and Grisogono, 2002).

Besides the pronounced cooling effect, relatively dry air, high wind speeds and enhanced turbulence characterise the kata-

batic wind layer over glaciers during the ablation season. A dry surface layer and a marked increase in humidity in the first tens



of metres above the glacier surface, such as observed on Kanderfirn (cf. Fig. 9), were also found on Pasterze and Mittivakkat Gletsjer (Van Den Broeke, 1997a; Hansche et al., 2023). Since the specific humidity on the Kanderfirn was higher throughout the sounded air column above the katabatic wind layer, it is very likely that the dry air near the glacier surface originates from the accumulation area, where evaporation is limited compared to the ablation zone, or from the free atmosphere above. The sat-

uration deficit caused by the dry low-level jet, together with relatively high wind speeds and increased turbulence, may favour evaporation and thus reduce the energy available for melting snow and ice in the ablation zone. This means that katabatic flow has the potential to reduce not only the sensible heat flux (e.g. Shaw et al., 2024), but also the latent heat flux compared to the microclimatic conditions (warmer and more humid air) outside the glacier. Since the temperature and humidity profiles observed above the top height of the surface-based inversions are linear (Figs. 6 and 9; Oerlemans et al., 1999), their extrapolation

to the glacier surface and deviation from the measurements below the inversion top height provide an alternative to off-glacier lapse rates and gradients (see e.g. Greuell and Böhm, 1998; Shea and Moore, 2010; Shaw et al., 2024) for estimating and parameterising the cooling and drying effect of the katabatic wind layer.

Above the katabatic wind layer, a well-developed mountain-valley wind circulation similar to that at Kanderfirn has been found at Pasterze and Vatnajökull during periods of stable weather (Van Den Broeke, 1997b; Oerlemans et al., 1999). A

characteristic feature is the decrease in wind speed from a maximum near the surface to a minimum about 100-200 m above the surface (cf. Fig. 11). At the height of the minimum wind speed, the horizontal wind direction changes from downglacier to upglacier (Fig. 12; Van Den Broeke, 1997b). The valley wind typically advects warmer and more humid air towards the glacier (Figs. 5 and 8; Van Den Broeke, 1997b). However, without additional atmospheric soundings along the glacier flow line, it is difficult to assess whether the glacier and valley winds are essentially decoupled during stable conditions such as on 16 June

2021, or whether entrainment of warm and humid air occurs at higher altitudes. To better capture the interactions between the different thermally driven winds in glacierised alpine terrain, parallel UAV-based atmospheric soundings at different locations on the glacier, together with ground-based measurements, would be helpful. Repeating such soundings under different synoptic conditions would help to identify the atmospheric conditions that favour the decay of the katabatic wind layer and the heat advection from outside the glacier. This information is crucial to improve the parameterisation of local atmospheric conditions

over mountain glaciers and to assess how a warming world affects the local circulation in alpine terrain.

## 6  Conclusions

With the UAV-based atmospheric sounding technique, we have presented a new approach to study the interaction of local winds in alpine terrain and investigate the structure of the atmospheric boundary layer over glaciers up to several hundred metres above the surface. The measurement technique provides a lightweight and low-cost alternative to tethered balloons and

complements ground-based measurements at weather stations and meteorological towers. Vertical profiles of air temperature, humidity, pressure, wind speed, wind direction and turbulence can be derived from the meteorological and flight recorder data collected by the developed open-source fixed-wing UAV. A drawback of the fixed-wing UAV is the extensive training that is required for safe operation in alpine terrain. However, a hybrid UAV combining efficient forward flight with hovering and

vertical take-off and landing capabilities is currently under development and will facilitate operation in alpine terrain. The
UAV-based atmospheric soundings conducted during the feasibility study at the Kanderfirn in the Swiss Alps add to the sparse
global dataset of atmospheric soundings in glacierised terrain and reveal typical features of the boundary layer over glaciers
in summer. A persistent low-level katabatic jet, characterised by a pronounced surface-based inversion, relatively dry air, high
wind speeds and enhanced turbulence, was observed at the Kanderfirn. Above the katabatic wind layer, a well-developed valley
wind adevecting warm and humid air from the periphery towards the glacier was found. While vertical profiles at one location
can provide fundamental insights into the structure of the boundary layer over glaciers, parallel UAV-based soundings at
different locations and repeated under different synoptic conditions would be desirable in the future to uncover the interactions
between the thermally driven local winds in alpine terrain and to assess the potential impact of rising off-glacier temperatures
on the katabatic wind and its wider cooling effect.

*Code and data availability.*  The atmospheric sounding and flight recorder data as well as the scripts for data post-processing, reformatting,
analysis and visualisation can be downloaded from the open-access repository Zenodo: https://doi.org/10.5281/zenodo.13889613 (Groos
et al., 2024). The mobile measurement post-processing (mmp) FORTRAN package can be downloaded from the following Git repository:
https://git.rz.uni-augsburg.de/philipan/mmp (Philipp, 2024).

*Author contributions.*  ARG conceived the study. ARG and AP designed the UAV. MB supported the development of the UAS and helped
to implement the Paparazzi UAV software and hardware. NB built the two UAVs with the help of ARG. NB and ARG carried out the mea-
surement campaign. AP developed the mmp software package for processing and analysing the flight data and atmospheric measurements.
ARG, NB and AP analysed the data. ARG drafted the manuscript and prepared the figures with contributions from NB and AP. All authors
contributed to the discussion of the results and revision of the manuscript.

*Competing interests.*  The authors declare that they have no competing interests.

*Acknowledgements.*  The expenses for the construction of the fixed-wing UAVs and the implementation of the measurement campaign on the
Kanderfirn were covered by the Institute of Geography of the University of Bern. We would like to thank Heinz Veit for the support of this
study and Peter Leiser for his help in soldering the electronic components of the two UAVs. Moreover, we wish to extend our gratitude to the
other lead developers of the Paparazzi UAV project (namely Gautier Hattenberger and Hector Garcia de Marina) for the fruitful discussions
and their support in implementing the UAV hardware and ground control station software.



# Appendix A: Sensor intercomparison

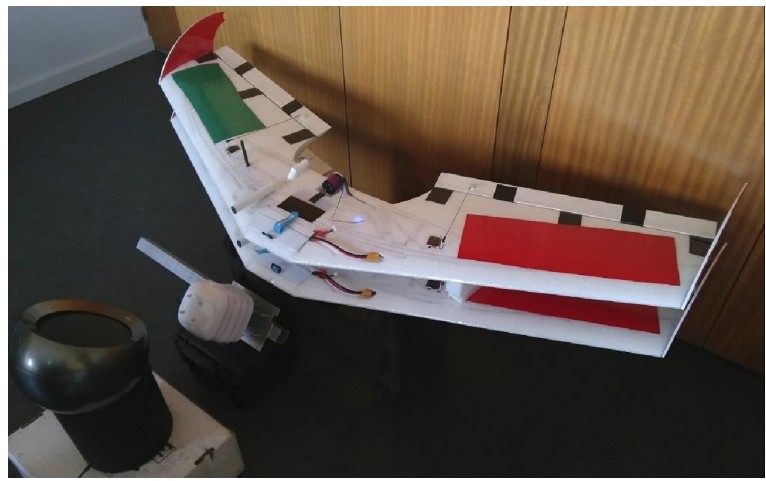

**Figure A1.** Setup of the 24-hour intercomparison measurement of the temperature and humidity sensors (SHT75) installed inside the white tube of each UAV and a reference sensor (SHT21, inside the radiation shield) placed in front of both UAVs. A fan ensured continuous air flow. The experiment was conducted from 4 to 5 June 2021.

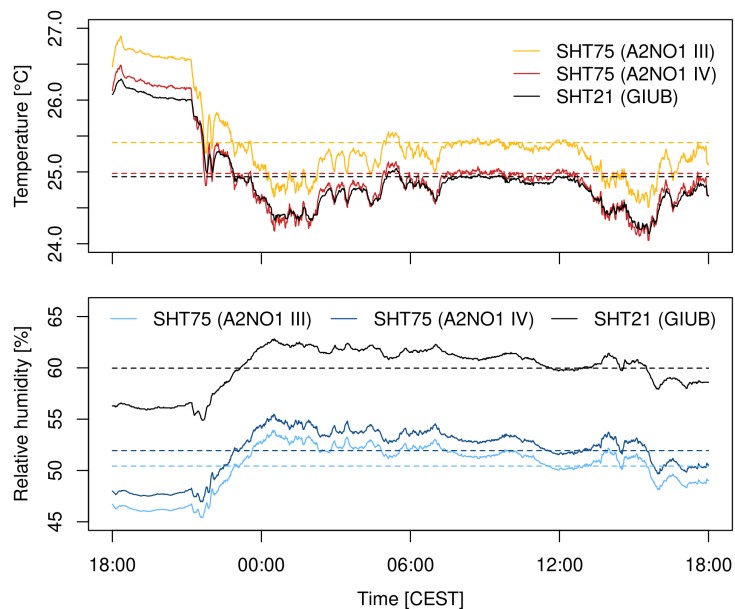

**Figure A2.** Result of the 24-hour intercomparison measurement. The mean difference in air temperature between both SHT75 sensors is 0.43 °C and the mean difference in relative humidity is 1.5 %.



## Appendix B: Lapse rates


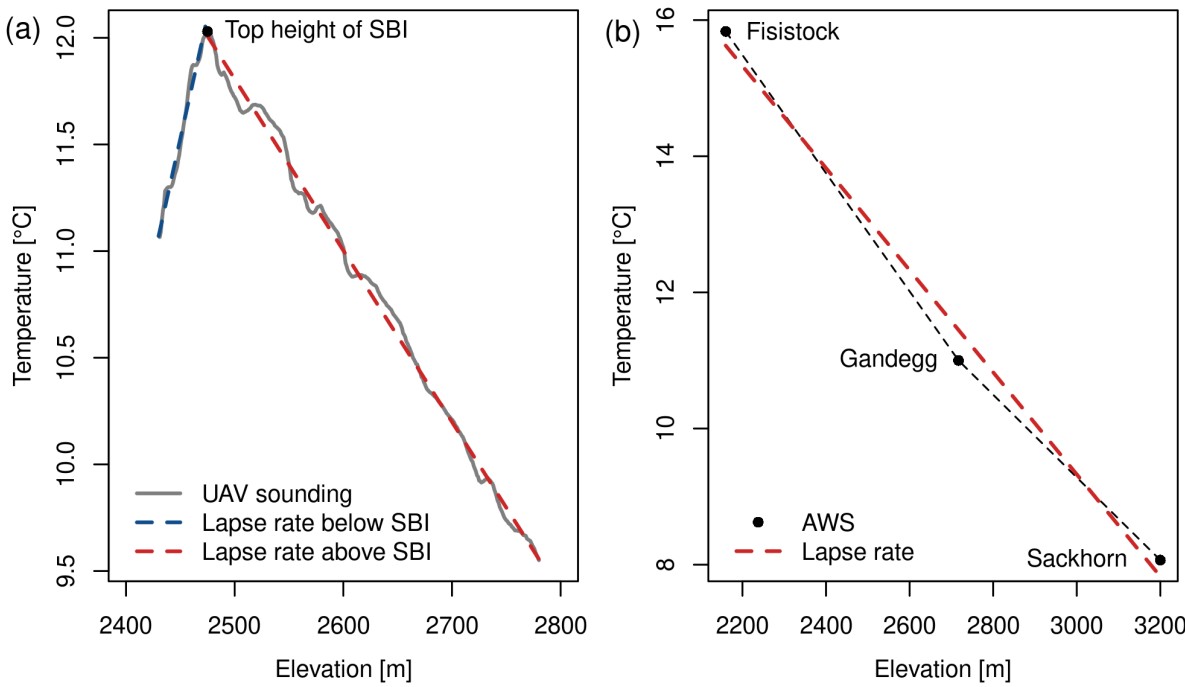

**Figure B1.** Example of the derivation of the lapse rate over the Kanderfirn and the surrounding area for 16 June 2021 at about 14:10. (a) Lapse rate below and above the top height of the surface-based inversion derived from the UAV-based atmospheric soundings. (b) Environmental lapse rate in the study area derived from the data of three nearby weather stations.





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
