# Peer review of "Atmospheric sounding of the boundary layer over alpine glaciers using fixed-wing UAVs"

_Atmospheric Measurement Techniques, 2024_

## Author Comment (AC1)

This is a combined response to the editor and both reviewers.

**General response**

We thank the editor, Maximilian Maahn, for overseeing the review process, and both anonymous reviewers for their constructive feedback on our manuscript. In the following pages, we address the reviewers' comments point by point. The reviewers' comments are highlighted in grey. We hope that our responses will warrant consideration of our revised manuscript.

**Response to the Handling Editor**

Please double check the journal's guidelines regarding figures, I noticed that subplot labels (a), (b), are missing for some plots and that units should be in round instead of square brackets.

We checked the journal's guidelines and made adjustments to the figures where necessary. We included the missing subplot labels and set the units in round brackets.

**Response to Referee Comment 1 (RC1)**

Review of "Atmospheric sounding of the boundary layer over alpine glaciers using fixed-wind UAVs" by Groos et al.

This manuscript describes the use of a fixed wing UAS to profile the lower atmosphere over a mountain glacier environment. Since data from only a single day is shown the results here are illustrative of the types of features that could be observed with a UAS field campaign but do not allow for any broader conclusions about glacial meteorology. While the results presented will be of interest as an illustration of the potential research applications of using a small UAS to study alpine glacier meteorology the presentation requires major revisions as described in the comments below. Once these major revisions are completed the manuscript will be suitable for publication in Atmospheric Measurement Techniques.

We appreciate that the reviewer supports publication of this manuscript in *Atmospheric Measurement Techniques* following the requested revisions. As stated in the title and abstract, the primary aim of the manuscript is to present a low-cost and open-source UAV-based atmospheric sounding technique (aligned with the scope of this journal) and to demonstrate its potential for investigating the atmospheric boundary layer over alpine glaciers. While we highlight various processes and features observable with this technique, we do not draw broader conclusions at this stage. To explore the influence of varying synoptic conditions and different surface types (snow, ice) on the structure and dynamics of the atmospheric boundary layer over the studied glacier, we incorporated data from three additional measurement campaigns during the ablation season 2021 (July, August, September), which include nocturnal flights and one extended sounding up to 800 m a.g.l. We now present the results from 40 instead of previously 8 soundings (79 instead of previously 16 profiles).

**Major comments**

A figure or series of figures illustrating the data processing described on pages 9 and 10 should be included to illustrate what the raw, unprocessed data from the UAS looks like and how that is modified prior to further scientific analysis. This figure(s) should show:

- unprocessed T profiles and the final smoothed profiles in 1 m bins.

- temperature bias between ascent / descent legs averaged to 1 m bins

- illustrate how lapse rates and inversion height are calculated by showing profile of T'(z). In particular I am interested in seeing how noisy the T'(z) profile is and what impact this has on identifying the height of the SBI.

- show RH, T and derived q profiles

- show profile or time series of original resolution roll rate and derived turbulence intensity proxy

These figures showing original data and derived data used in the results section will allow the reader to clearly see how the data was modified to allow for subsequent scientific analysis.

As suggested, we have prepared additional figures for the manuscript and appendix that present both the original and processed data, as well as the effects of the applied modifications on the subsequent scientific analysis. The data have now been aggregated into 5 m vertical bins.

I found the color shaded time-height plots to be attractive but ultimately not very useful for understanding the features present in the UAS observations. I strongly suggest that the authors replace these figures with single plots for each each variable (T, q, wind speed) showing all descent profiles from all flights. By showing all of the profiles on a single plot it will make it easier to see details in the change in magnitude and vertical structure over the course of the day than the color shaded cross-sections currently shown. To help interpret the time evolution shown in this plot each descent profile should be shown in a different color (maybe ranging from blue to red with increasing time of day).

We agree that additional figures displaying all profiles from a single campaign in one plot for each variable are helpful for visualising changes in magnitude and vertical structure throughout the day. However, we have chosen to retain the heat maps for *air temperature* and *specific humidity* as an alternative, intuitive visualisation of spatio-temporal variations in the boundary layer.

Showing profiles of wind direction, in addition to the wind roses shown in Figure 12, would make it easier for the reader to see the relationship between the switch from down glacier to up valley wind direction and differences in speed.

We have included additional figures that present wind speed and direction profiles in the same plot to facilitate interpretation and discussion.

It would be useful to show a synthesis plot at a representative time showing profiles of all of the analyzed variables together to illustrate how the different variables and their profiles relate to

each other.

We have added a couple of synthesis plots at selected times of interest to illustrate the relationships between the different variables.

What is the explanation for the nearly linear lapse rate for profile 1 down to the lowest observed height in the 16:05 sounding in Figure 6? This differs from all of the other profiles and is markedly different from the profiles at adjacent times. Is this an observational error or a real feature of the atmosphere?

We have no evidence that the discrepancy between the two profiles in the 16:05 sounding (original Fig. 6) and the absence of a low-level inversion in Profile 1 is due to a measurement error. It is likely that we observed a temporary erosion of the glacier wind and advection of warm air from the surrounding slopes or glacier forefield at the time of Profile 1. However, we cannot provide further evidence to support this hypothesis, as we were not operating a weather station on the glacier during the ablation season in 2021.

Uncertainty in the observed quantities and the impact on interpretation of the results needs to be included. In particular, what is the uncertainty in the derived wind speed and direction and does this alter the interpretation of the results. In particular I am wondering about the rapid shift in wind direction and how this is handled if the spiral path used to calculate wind speed and direction spans both down and up valley wind directions. Does this account for the low wind speed at the height of the change in wind direction (i.e. it is an artifact of how the wind is derived rather than a true feature of the wind profile?).

We agree that providing uncertainty estimates is important; however, in this case, they are difficult to quantify precisely. We have modified the methodology to include a measure of uncertainty. Specifically, we now calculate the mean and standard deviation for each 5 m bin based on the two descent profiles and the second ascent profile of each sounding (all time-lag corrected). The standard deviation reflects both the measurement uncertainty and the short-term variability between subsequent profiles within a single sounding. Where it did not overly clutter the figures, we have included the standard deviation as a semi-transparent shading around the profile lines.

Mayer et al. (2012), who used a similar approach to estimate wind speed and direction from ground speed and flight path azimuth data obtained from the autopilot's GPS system, reported good agreement between their wind profiles and measurements from conventional atmospheric profiling systems such as radiosondes and tethered balloons. As we applied a low sink rate and performed spirals with a high number of cycles, we assume that the algorithm yields reliable results. To further assess the uncertainties associated with this wind profiling technique, a direct comparison with other instruments (e.g., lidar, sodar, RASS) is planned for this summer on an alpine glacier.

The distinct change in wind speed observed is characteristic of a wind shear layer, where a natural reduction in wind speed occurs. This feature is not an artifact, but rather a consistent representation of the actual atmospheric state.

Minor comment

Lines 110, 123: Figure 2 should be figure 3

Thank you for the hint; the correction has been made.

**Response to Referee Comment 1 (RC2)**

The manuscript presents one day of profile measurements (8 individual measurement flights with profiles each to an altitude of max. 400 m above the ground) with a fixed wing UAS over a Glacier surface. One of the stated main goals of the study is the characterization and investigation of the structure and development of the katabatic jet over a glacier, a phenomenon that typically extents only a few tens of meters above the surface, and can by that not appropriately probed by profile measurements with a fixed wing UAS. Here multi-rotor drone systems (preferably with forced ventilation of the temperature and humidity sensors) with their capability of hovering and ascending/descending very slowly, would be the by far better choice. The manuscript covers a lot of different topics (a bit of system description, a bit of methodology, a bit of scientific evaluation), but is not going deep in any of them. The most interesting and novel part of the study, the estimation of a turbulence proxy, is again only touched and not described (it is referred to a separate publication). Integrating that part in the presented manuscript would clearly help to make it publishable, in its present form it is too thin and lacking novelty.

However, I see a clear potential of combining the presented one day of measurements with additional measurements in comparable environments, or as mentioned above include the detailed description of the turbulent proxy algorithm.

The primary aim of this manuscript is to present a low-cost, open-source UAV-based atmospheric sounding technique and to demonstrate its potential for investigating the atmospheric boundary layer over alpine glaciers. This focus aligns with the scope of *Atmospheric Measurement Techniques* and also explains why the scientific analysis has not been more extensive at this stage. The characterisation and investigation of the katabatic jet over the glacier represents only one aspect of the study, which may have been overemphasized in the introduction (this has been adjusted accordingly).

Although the low-level jet reaches indeed its maximum wind speed within a few metres above the glacier surface, the air temperature and specific humidity profiles clearly show that the katabatic wind influences the boundary layer structure above the glacier up to 50 m or more. Previous studies have similarly reported that the cooling effect of glacier winds can extend up to 100 m (e.g., Van den Broeke, 1997a,b). The temperature and humidity profiles clearly demonstrate that the low-level jet is captured by the fixed-wing UAV soundings. Therefore, we do not understand how the reviewer concluded that this phenomenon "can by that not appropriately probed by profile measurements with a fixed wing UAS". However, we agree that a rotary-wing UAV might be benefitial for the soundings of the lowest metres and that both types of UAVs have their respective advantages and disadvantages and should therefore be combined in future campaigns

(we discuss these aspects in more detail in the revised manuscript).

Additional measurement campaigns were conducted during the 2021 ablation season (July, August, September), including nocturnal soundings, to investigate the effects of varying synoptic conditions and different surface types (snow, ice) on structure and dynamics of the boundary layer over the glacier. Initially, we chose not to include these data in order to maintain a clear focus on the measurement technique itself, its potential, and its limitations. However, based on the valuable feedback from both reviewers, we have decided to present and discuss the data from all four campaigns in the revised manuscript.

The novel contribution of this study lies in the first application of a UAV for boundary layer research on a mountain glacier – a particularly challenging environment – and the proof of concept of the developed lightweight, low-cost and open-source UAV-based atmospheric measurement technique. Moreover, the open-source nature of the system allows for the direct integration of additional sensors and instruments, which is a key advantage over closed (commercial) systems that often lack interfaces for sensor integration and hinder the combined analysis of flight recorder and observational data.

Some additional comments that hopefully can support a new submission of the manuscript:

Introduction: there is a vast discrepancy between the identified scientific gaps and goals and the very limited measurement capabilities of the presented/used UAS; as mentioned in the general comments fixed wing is not appropriate for the shallow katabatic jet; and the ceiling altitude of max. 400 m is far from sufficient to investigate the larger scale valley wind systems in the study area

As outlined above and demonstrated throughout the manuscript, the presented fixed-wing UAV is capable of profiling the boundary layer above a mountain glacier. The wind speed profiles show that the valley wind can be detected. The maximum altitude reached during the campaigns (400 m a.g.l.) does not represent the technical limit of the fixed-wing UAV. We have included an example from the August campaign demonstrating that the system is capable of conducting soundings up to 800 m a.g.l. and beyond.

line 138: "the sink rate was low to minimize....."; you should quantify this; I would even suggest that you also plot the profiles of vertical velocity for all descents to get a feeling how constant this is and which effect it could have on the measurements due to the sensor time constant data processing and analysis: You state that met data and flight data are collected/stored separately. Are those data sets in any way synchronized by a common clock/timer?

The mean sink rate for most profiles was on the order of 1–2 m/s. We have added a figure in the appendix displaying the vertical velocity for all profiles. A time-lag correction was applied to all profiles to account for sensor inertia (see revised methodology section).

While meteorological and flight recorder data are stored in two separate files, all data are consistently synchronised using the autopilot's clock.

Line 161 and Appendix A1: you apply a simple shift in temperature (of about 0,4 C) that you

gained from one day of parallel measurements at an ambient temperature of around 25 C; there should be at least a second comparison been done at a distinctly (ideally close to operational conditions over the glacier) lower temperature to check for potential gain errors, that would cause a temperature dependent change of the differences.

Yes, a direct intercomparison on the glacier would have been ideal, but this was not feasible during the campaigns due to time constraints. However, since nearly all soundings within each campaign were performed using the same UAV (and thus the same sensor), this bias does not have practical implications for the results presented.

Figures 6/9/11: it would be so great to have a ground value (e.g. by a simple weather station placed close to the UAS start/landing site) to verify/validate the very strong gradients that often occur close to the ground in your observations

We fully agree. However, due to limited personnel and funding, we were unable to operate a weather station on the glacier during the 2021 measurement campaigns. Nevertheless, strong temperature gradients and surface-based inversions have been documented in previous studies and are characteristic features of glacier winds (e.g., Van den Broeke, 1997a,b; Mott et al., 2020). During "The Second HinterEisFerner EXperiment" (HEFEX II) (Nicholson et al., in re-review for *BAMS*), similar UAV-based soundings were conducted over multiple on-glacier weather stations, confirming the strong gradients observed in the present study.

Figure 12: it feels inconsistent to present the wind information as wind roses, while all other

parameters are given as profiles

We have included additional figures that present wind speed and direction profiles in the same plot to facilitate interpretation and discussion.

References: Groos et al (line 429); inconsistency in abbreviation of journal name

Thank you for the hint, we have corrected this.

---

## Author Response (AR2)

**General response**

We would like to thank the editor Max Maahn and both anonymous reviewers for providing swift feedback on the revised version of the manuscript.

**Response to the Handling Editor**

Dear authors, thank you for revising your manuscript. Please consider the comments by the reviewer before I can accept the paper for publication. Regards, Max Maahn

We appreciate the swift handling of the review process, and have addressed the two minor comments from RC2 in the revised version of the manuscript. We have also corrected a few typos and linguistic imprecision that we discovered in the meantime.

**Response to Referee Comment 1 (RC1)**

The authors have adequately addressed all of the comments in my review of the initial version of this manuscript and I now find this manuscript suitable for publication in Atmospheric Measurement Techniques.

We are pleased that the reviewer is satisfied with the revision and now recommends publication.

**Response to Referee Comment 1 (RC2)**

The manuscript has improved considerably after incorporating many of the reviewers' comments. I can accept it for publication after the incorporation of two minor comments/technical corrections:

- 1) Fig. 7: what is the reason to use different colors for the same profile in the different parameters (T, q, ws, wd). It makes it just unnecessary complicated to compare the conditions at the same time. I recommend strongly to change this in the final version.
- 2) line 230: With respect to the turbulence proxy "Details on this method are published in a separate paper."; if published, please include the reference; if not yet published, please modify the sentence correspondingly

We are pleased that the reviewer sees a significant improvement in the manuscript and now recommends publication.

- 1) We used different colour schemes to help the reader to distinguish between the four variables, but we agree that one common colour scheme facilitates comparison of the variables at the same sounding time. Hence, we modified Fig. 7 accordingly.
- 2) Since the separate manuscript on the turbulence proxy methodology has not yet been published, we deleted this setence.